# Development in Sustainable Concrete with the Replacement of Fume Dust and Slag from the Steel Industry

**DOI:** 10.3390/ma15175980

**Published:** 2022-08-29

**Authors:** Maria Eugenia Parron-Rubio, Benaissa Kissi, Francisca Perez-García, Maria Dolores Rubio-Cintas

**Affiliations:** 1Department of Civil and Materials Engineering, University of Malaga, C/Doctor Ortiz Ramos, s/n, 29071 Malaga, Spain; 2ENSAM, Mechanical Engineering Department, Hassan II University, Casablanca 20000, Morocco; 3Department of Industrial and Civil Engineering, Campus Bahía de Algeciras, University of Cádiz (UCA), Avda. Ramón Puyol, s/n, 11201 Algeciras, Spain

**Keywords:** electric arc furnace, stainless steel fume dust and slag, cement replacement, resistance and leaching

## Abstract

Nowadays, the reuse of waste is a challenge that every country in the world is facing in order to preserve the planet and introduce a circular economy. The chemical composition of some steel waste suggests that there are potentially appropriate substances for reuse, since this type of slag undergoes a process similar to that of cement in its manufacture. The advantages for the environment are obvious, as it valorises waste that is deposited in landfills. This paper studies the field of stainless steel, because its composition is different from that of carbon steel, and the replacement of cement with material or waste produced in the manufacture of stainless steel in a concrete matrix. This paper presents the results of replacing 25% of cement with material or waste produced in the manufacture of stainless steel in a concrete matrix whose values in the substitutions carried out were around 21% and 25% in terms of increased resistance capacity. These results have been obtained by carrying out tests, in terms of both strength and environmental capacity, allowing us to determine viable applications for the use of steel waste to improve the performance of cement or at least match it.

## 1. Introduction

Industry today generates quantities of waste in both manufacturing processes and different types of production during the material conformation process. It is possible to reuse this waste instead of taking it to a landfill, where in many cases the process of treatment and deterioration is very slow, with the consequent deterioration of the environment [1,2]. A thorough understanding of the potential effects of waste, whether urban or industrial, will facilitate environmental protection and, in turn, add a fundamental economic and social dimension to the circular economy and environmental health that will allow us to strike a balance between sustainable development and the environmental interests of future generations [3]. Among various studies, we can find the durability [4] and resistance properties of concrete with different compositions and dosages, in which certain results are obtained that may be of great importance [5,6].

Various industries, mainly the steel industry, generate large amounts of waste that are taken to landfills. The process of obtaining steel involves a considerable amount of highly toxic waste due to the heavy metals it contains. Fume dust and slag dust are industrial waste produced by the iron and steel industry during the steel smelting process. The process of manufacturing stainless steel at steelworks with an electric arc furnace (EAF) from a mixture of scrap and ferroalloys produces many gases captured by various systems. The dust retained in these filters passes to silos to be collected in bags. This fume dust is a mixture of oxides, mainly iron and chromium.

From the environmental point of view, the current legislation considers fume dust to be toxic and dangerous due to the ease of the leaching of the species it contains and its association with elements such as lead, zinc, and chromium. On the other hand, even though the slag is not toxic, it must also be taken to landfills due to its high basic composition [7]. In addition, stabilising this type of material before taking it to a landfill has been studied [8,9,10].

One solution to this problem, for both fume dust and slag, is to incorporate it into civil engineering, reusing waste that would go to the landfill.

Several studies are currently available about the possible use of steel slags as an addition or cement replacement in concrete [11]. Moreover, research has been carried out with the aim to assess the influence of using coarse slags as aggregates in concrete [12]. Similarly, blast furnace slag has been considered as a possible replacement for cement, and for some mixtures and replacement ratios, an increase in concrete strength has been observed [13,14,15,16]. Conversely, blast furnace slags are also employed as a replacement for aggregates and fly ash in concrete production [17,18,19,20,21].

However, it is worth highlighting that slags resulting from blast furnace (BF) and electric arc furnace (EAC) slags have different properties.

Therefore, the present study focuses on the latter, which is utilised in the production of stainless steel. Various studies have demonstrated the potential of ferritic smoke powder as an “addition” to a reference concrete mixture [22]: enhanced mechanical properties, such as compression strength and Young modulus, have been observed in some cases [23].

The present research aims at understanding the influence of “replacing” part of Ordinary Portland Cement (OPC) with various by-products deriving from the production of stainless steel in EAF. Specifically, austenitic fume dust (FDA), ferritic fume dust (FDF), and treated slag (TS) are considered in the present study as a cement replacement, which is one of its main novelties.

First of all, a thorough physical and chemical characterisation of the by-products under consideration is proposed in Section 2. Specifically, thermogravimetry analysis is employed for fume dust, whereas sieving grade characterisation is carried out for slags. Then, starting with a reference mixture, 25% in weight of OPC is replaced by either fume dust or slag collected in EAF plants. This percentage was selected on the basis of the results of previous studies [15]. The concrete mixtures made with the aforementioned by-products in the partial replacement of OPC are presented and the results obtained from experimental tests are reported and analysed in Section 3. The main findings are outlined in Section 4 and pave the way for a new prospect for implementing circular economy practices in the stainless steel industry.

## 2. Experimental Work

### 2.1. Materials

In order to avoid pre-existing defects in the concrete, it is proposed in this work to study its components using behaviour models that give a better understanding of dosing. This is how the characterisation of fume dust is analysed, as a product coming from the combustion of steel in an electric arc furnace, which is collected in bags and then taken to special landfills due to its metal content, such as nickel, chromium, manganese, etc. Slag is produced in the steel skimming process and undergoes a different cooling and refining process than flue dust.

Figure 1 shows the aspects of each type of slag and fume dust. The fume dust colour is due to different additions to the steel.

The intention is to cross the inductive method, with tests of limited validity and fixed to the number of tests carried out, with the deductive method, where the microstructure is identified to obtain, based on modifications within it, materials with macroscopic properties to measure.

The solution will lie in achieving a scale that allows taking advantage of the macroscopic formulation after averaging the typical microstructural characteristics of the material and its components.

The advances in the research framework in terms of modelling and analysing the behaviour of the structural systems of cementitious matrices suggest the need to develop research work, beginning with taking samples of different elements that will be used for testing.

### 2.2. Morphology

Fume dust is made up of spheroidal particles of highly variable diameters, ranging from 50 to 350 microns. It is a very powdery material with a moisture content of less than 1% by weight. In order to study its composition and granulometry, samples were taken from a random fraction contained in bag filters where the residues of ferritic and austenitic fume dust are stored. For this study, 20 kg samples of the compounds were taken and stored in closed containers. For each sample, 250 g was studied, with a granulometric study performed as shown in Figure 2. A magnetic laboratory sieve shaker and sieves with 8-inch diameters and a mesh size between 1 and 0.038 mm were used for this purpose. In all, 77.06% of the grains were within this range. It was observed that there was no symmetrical form in the particle distribution, but 19.17% of the material analysed tended to the right and 3.77% to the left.

In order to study the chemical composition of waste, it is necessary to identify all of the components that comprise it. The data on the composition of metallic oxides or metals in these residues of dust and slag were provided by the company that manufactured the material and are broken down in Table 1 and Table 2.

Once the composition of the materials was obtained, the thermogravimetric analysis of the materials was carried out. This is a technique in which a certain mass of a substance, on the order of a few milligrams, is monitored as a function of temperature or time when subjected to a temperature-controlled program. The results obtained are shown in thermograms of the fume dust samples (Figure 3).

X-ray diffraction and thermogravimetry allow knowing the contents of 19 different species of chemical elements. First, X-ray diffractometry gives structural information about a sample by determining the crystalline species in said sample. The thermogravimetric analysis presents a curve, in which the temporal derivative of mass loss as a function of time or temperature is observed.

In Figure 4, the diffractometric analysis shows a complex material constituted fundamentally by metallic oxides with the general formula XY_2_O_4_, where X is a divalent metal (magnesium, nickel, or iron) and Y is a trivalent metal (iron or chromium) in a spinel-type structure with more or fewer variations. The majority species detected in the diffractograms was iron chromite, with the formula FeCr_2_O_4_, presenting an inverse spinel structure. This species is of great interest for ceramic development, due to the behaviour of the ternary system Cr_2_O_3_–FeO–MgO.

The following reaction was observed at around 1000 °C:FeCr_2_O_4_ + MgO → MgCr_2_O_4_ + FeO(1)

Due to the similarity of the crystal structure and lattice parameters of the different ferrites and chromites, an exact peak assignment is difficult. In this way, it would be possible to assign other species of the same type, such as zinc ferrite (franklinite) and magnesium chromite. Furthermore, we have to assume the existence of species associated with silica/lime/calcium silicate, whose amorphous structure prevents their detection, thus justifying the silicon and calcium contents obtained in the chemical analysis [22].

The thermal analysis shows an initial zone of continuous loss, assignable in principle to the dehydration process of the absorption/constitution water that usually exists in materials that present fine granulometry. However, this loss coincides with the first peak of the carbon and moisture graphs. This led us to suppose that in this zone (below 400 °C), decomposition that produces carbon and water, possibly assignable to organic carbon from oily products, also takes place. At around 400 °C there is a marked loss, probably due to the dehydration process of free calcium hydroxide, which coincides with the second peak of the water graph.

On the contrary, as shown in the thermogram, the process was interrupted at around 450 °C, with an observed gain zone, which could be associated with an oxidation phenomenon of some spice or with structural transformation. This phenomenon happens again at 600 °C and could be some other transformation of maghemite to hematite. In the thermogram, these phenomena mask the losses due to the decomposition of calcium carbon, which can be observed in the second peak of the carbon graph, also giving us information about the decomposition of alkaline carbonates at a temperature of 850 °C.

### 2.3. Description of the Type of Cement, Slag and Sand Used

The cement used to develop the specimens was ordinary Portland cement with the following characteristics: CEM I 52.5 R type with a density of 3.1 g/cm^3^. The components of the cement were according to the manufacturer. This type of cement was used because it does not have any type of addition that would mask the results obtained.

The aggregate used was crushed limestone from commercial manufacturing plants located in the Campo de Gibraltar. The proportion of aggregate used in this work was 50% 0–2 and 0–4 mm sand and 50% 0–16 mm gravel. As shown in Figure 5, the granulometry of each aggregate used was according to UNE-EN 933-2 [23].

### 2.4. Additive

In order to meet the workability requirements of the concrete, a superplasticiser additive was used: Glenium ACE-324, manufactured by BASF. Domestic tap water was used.

### 2.5. Description of the Process

Regarding the percentage of substitution in cement, in previous studies with different residues from steel mills [10,24,25,26], tests were carried out with substitution percentages ranging from 5% to 35%, and the most representative values were found between 25% and 27%. Therefore, in order to homogenise the process, 25% was chosen, since it was the average of the best results of all tests carried out.

The procedure for the elaboration of concrete comprised 3 daily mixes for each substitution that was carried out. At the same time, in order to have a comparison for each mix, another mix of conventional concrete was made, all with the same materials, so that there would be no interference. In all cases, 25% of the cement was replaced with fume dust or slag (Table 3).

In previous studies carried out with different residues [13,24,25], for this type of residue, it was concluded that the most optimal proportion was 25%. In previous tests, substitution between 27% and 30% gave optimal results for some types of slag and fume dust. Therefore, in order to homogenise the process, 25% was chosen, since it was the average with the best results of all tests carried out. The materials were as follows: concrete without addition (OPC), austenitic fume dust (AFD), ferritic fume dust (FFD), and treated slag (TS).

Once the concrete mixing process was finished, the moulds were filled and vibrated on a table for compaction at a frequency of 42 Hz (2400 cycles per minute) according to UNE 12390-2 [27]. Then, they were placed in a humid chamber for curing, as this is an enclosure that allows the interior to be maintained at relative humidity equal to or greater than 95% and a temperature of 20 ± 2 °C (UNE-EN 12390-4) [28].

### 2.6. Workability

The slump test was used to assess the workability of all fresh concrete mixes, adhering to EN 12350-2:2021 [29], to obtain within the settlement test of the fresh concrete a plastic consistency between the values 60 and 90 mm.

### 2.7. Geometry of Specimens

The geometry of the specimens was standardised according to EN-12390-1 [30] as cubic with a main dimension of d = 100 mm, based on the measure having to be at least three and a half times the nominal size of the aggregate in concrete.

The moulds in which the tests were carried out were watertight and non-absorbent.

Flexural tests with the same characteristics were carried out with prismatic specimens with dimensions according to the standard of a square with edge d and length 2 d or 4 d; in our case, the dimensions were 40 mm × 40 mm× 160 mm.

## 3. Result and Discussion

### 3.1. Workability

For the study of the workability of the mixtures, the average of each kneading was taken. It was observed (Figure 6) that the workability of mixtures AFD and FFD decreased by approximately 7.1% and 1.5% with respect to OPC, which was the more fluid sample in both cases. On the contrary, in TS, workability decreased by 21.5% with respect to the control concrete [6,31].

This may be due to the fact that the compositions of AFD and FFD do not absorb as much water, as the granulometry is smaller; therefore, fine particles can enter the interstitial spaces, preventing the generation of voids. TS, on the other hand, had a slightly larger particle size and was considered to have a smaller distribution at the spatial interface.

### 3.2. Water Absorption

These tests were carried out according to the UNE 83982:2008 [32] standard for the determination of water absorption by capillarity in hardened concrete. As observed in Figure 7, water absorption is related to workability. Both the maximum and average absorption are higher in OPC than in the mixes with fume dust and slag. In AFD and FFD, the maximum water absorption was 18% and 23% and the average water absorption was 36% and 37.8%, respectively. The values for ST were 10.4% and 5.6% with respect to the control. Absorption indicates the amount of water used by the material to fill the pores of the concrete, so this shows that the material with substitution had decreased porosity, with stronger microstructural compaction than the control [33]. When substituting TS for cement, it was found that porosity was greater, more closely resembling OPC, while AFD and FFD showed decreased absorption, which is consistent with workability, since the cement and the fume dust change together in a pozzolanic reaction by the silica contained in the residue. Therefore, AFD and FFD have greater durability due to the lower absorption of water in the dry concrete, translating into increased resistance, as described in the following section.

### 3.3. Uniaxial Compression Tests

In order to carry out this test, six specimens were used for each mix manufactured per day. The results obtained are the mean of breaking values with a deviation of ±5 MPa, using an average of two specimens for each rupture, all of them remaining within the tolerance of ±5 MPa according to regulations; one more specimen must be broken if included. in the values. The specimens were broken at 7, 28, and 90 days using a hydraulic press formed by two perfectly rigid compression plates. The upper plate was linked by an articulated joint. This joint retains the load on the specimen even if there is a defect between the flat surfaces.

Regarding the applied load, a constant load speed of 0.5 MPa/s was selected (N/mm^2^s).

Table 4 shows the test results of the cubic specimens and the mean values of the stresses obtained; two from each batch were tested. This test was carried out according to UNE-EN 12390-3 [34].

The results obtained show that at 7 days, the materials reached almost 75% of the strength capacity. The concrete with FFD and AFD substitution was at approximately 20.5% and 20%, respectively, with an increase in resistance capacity observed with respect to the control. Regarding TS, there was a 21% loss of resistance with respect to the conventional mix. This loss may occur because the reaction of the slag increases the chemical shrinkage of calcium silicate hydrate (C-H-S), which leads to an increase in microcracks and reduced resistance [35,36]. It was observed that in AFD at 28 days, the increased strength was maintained at a rate of 12.4% and in FFD at 21.7% These increases may be due to the fineness of the material and the absorption of water by the residue [10,20,21] hydrating the cement and reducing the air content in the interstitial spaces. As it has practically the same fineness as the cement, the internal reaction produced is favourable to the mixture. The slag, although it had increased resistance capacity, practically remained constant and did not increase excessively with respect to the rest of the waste. In relation to Young’s modulus, the tests were carried out according to UNE-EN12390-13:2022 [37], the specimens used in this case are cylindrical at 15 cm × 30 cm in order to carry out the test, and show linear behaviour depending on the resistance capacity of the different dosages of treated slag, ranging from 28.6% to 42.3% in FFD. It was observed that the material continued to have increased resistance capacity at 90 days, 11% for AFD and 13.6% for FFD, and with time, the increase continued with respect to OPC. This is consistent with the workability and absorption of the material.

### 3.4. Flexural Strength of Specimens

Flexural tensile tests were carried out at 28 days in accordance with UNE-EN 12390-5:2021 [38]. For each test, six specimens were used, with a deviation of ±5 in the results obtained. These are shown in Table 5.

Regarding the control mix, at 28 days it was observed that the flexural strength in all substitutions was practically the same in the three dosages studied. A 14.5% increase was observed in TS with respect to OPC, 14.67% in AFD, and 14% in FFD. This may be because slag and fume dust react, forming stronger particles in the bond, and when these materials take the place of internal air, much less cracking is produced.

### 3.5. Scanning Electron Microscopy Study

The four samples to be studied were prepared for sweep samples. Examining them by scanning electron microscopy required, first of all, eliminating the moisture content, since the working conditions of the microscope require a high vacuum. To do this, each sample was heated in an oven at about 100 °C for at least 2 h. The sample was then cleaned in an argon–oxygen plasma to remove carbonaceous residue from the surface.

The SEM results for OPC are shown in Figure 8.

The SEM results for austenitic fume powder are shown in Figure 9.

The SEM results of ferritic material are shown in Figure 10.

The SEM results of slag are shown in Figure 11.

It was observed in practically all tests that there was shrinkage cracking, which partly justifies the resistance obtained previously.

In Figure 9, Figure 10 and Figure 11 and the X-ray map in Figure 12, OPC appears to have mainly intense signals of oxygen, silicon, and calcium, indicating the presence of many traces of silica and calcium silicates in the concrete, which are sometimes associated with magnesium, aluminium, and potassium, as indicated by their X-ray maps.

Figure 9 shows austenitic fume dust, in which crystals of different shapes and sizes are observed, as well as areas of different depths. As in OPC, there are mainly intense signs of oxygen, silicon, and calcium, indicating the presence of calcium silicates in the concrete, which sometimes appear associated with magnesium, aluminium, phosphorus, and potassium (Figure 12), as the X-ray map shows [39].

Figure 10 shows large crystals. As in the previous images, intense signs of oxygen, silicon and calcium appear, indicating the presence of a multitude of silica and calcium silicate residues in the concrete, which sometimes appear associated with magnesium, aluminium, potassium and zinc, as indicated by the X-ray maps (Figure 12). The presence of residues rich in titanium is observed.

Figure 11 shows crystals of different shapes and sizes, as well as zones of different depths. In the X-ray maps of the elements identified in the represented area, as in the previous ones, there are intense signs of oxygen, silicon, and calcium, as well as traces of silica and calcium silicates in the concrete, which sometimes appear associated with magnesium, aluminium, and potassium, as they appear in the X-ray map (Figure 12).

### 3.6. Leachate Analysis

Fume dust is a material obtained during combustion in the steel manufacturing process. Thus, it contains heavy metals such as total chromium, nickel, etc., in its composition. According to regulations, it is considered a toxic material, so leaching it once it is encapsulated is of paramount importance [40,41].

On the other hand, slag is considered to be a nontoxic material, but as mentioned above, it must be deposited in a landfill due to its high basicity.

The distribution of components encapsulated in a cementitious matrix resembles the behaviour of a leaching column, in which the soluble constituents descend by gravity in the profile. Percolation water is the most active agent in the development of the soil profile. It is also of great interest to study the degree of fixation or mobility of certain fundamental components in the development of biomass [42].

The leaching process is used in studies to predict the behaviour of toxic or dangerous species present in solid waste that were subjected to various types of chemical attacks as a means of recovering them so that the waste does not exceed the maximum concentrations required by environmental regulations.

The composition of the aqueous phase in hydrated cementitious materials gives us important insight into the solid and liquid phase iteration processes within the chemical processes [42].

In the development of the leaching process, the specimen was introduced into 800 mL of distilled water for 24 h. The test was repeated by immersing the specimen for 24 h in another 800 mL of water, repeating the described methodology. After the test period, the sample was extracted and the leach solution was transferred to a 1000 mL volumetric flask filled with distilled water. Subsequently, the test tubes were artificially aged for 3 months and the same process was carried out.

The results obtained are shown in Table 6.

A Directive of the Official Journal of the European Communities, OJ L 11, 16.1.2003, and a Directive of the Council Decision of 19/12/2002, 2003/33/EC, address criteria and acceptance procedures for waste in landfills.

According to this standard, the maximum limit for chromium is 4 mg/kg. By balancing the total chromium and chromium IV from Table 6 in mg/kg dry matter with an easy ratio, it was determined that the leaching samples were lower in all material substitutions.

Taking 2 L/kg for comparison, the value reaches 0.315 mg/kg, well below the maximum limit of 4 mg/kg imposed by the standard.

The variability of other metals is qualitatively large but quantitatively insignificant, with the narrowest previously noted in Cr, which represents one-tenth of the assigned limit. In addition to the leaching limit values mentioned in Table 6, granular residues must comply with an additional criterion, pH ≤ 6.

The lowest value recorded in the analysis is 8.156, which corresponds to OPC, and the remaining values were higher.

The rest of the metals were not taken into account, since their proportions were low and they are not harmful.

## 4. Conclusions

The main objective of the analysis was to study the mechanical resistance and durability properties of concrete with the addition of fume dust from electric arc furnace steelworks and slag from various steel production processes. The main result is increased resistance in both compression and flexure provided by the residues added as components to conventional concrete.

The water absorption was lower by 23% and 37.8% in AFD and FFD, respectively, and by 5.6% in TS with respect to OPC, indicating that the pores decreased with the addition of AFD and FFD and remained practically the same in TS.The 20.5% increase in the compressive strength of concrete manufactured with AFD added and the 20% increase in FFD at 7 days maintained the same progression with respect to conventional concrete, with 12.3% in AFD and 21.7% in FFD at 28 days. At 90 days, hardening was 11% in AFD and 11.3% in FFD. Young’s modulus reached a range proportional to the resistance capacity of the addition.Regarding the resistance to flexotraction, the increase was practically equal to 64% in all tests, including the slag. A greater short-term hardening of the concrete made with both fume dust and slag was found, with around a 25% improvement in resistant behaviour. The data verify the success of the proposed experimental model.Steel residue at a substitution of 25% by weight of cement can solidify and stabilise safely, and it does not represent a threat to the environment, as the values of Cu, Pb, and Fe are below the allowed limits.

Concrete elements with superior mechanical properties and performance can be produced by using the waste of stainless steel instead of cement. Thus, new solutions were provided for the environmental problems originating from the steel sector.

It is understood that it is possible to obtain products with the performance specified in the related product standards by using EAF waste stainless steel in civil engineering. These results provide a better indication as to the possibility of using EAF fume dust in other aspects of industrial/civil engineering by recovering and reusing this residue.

## Figures and Tables

**Figure 1 materials-15-05980-f001:**
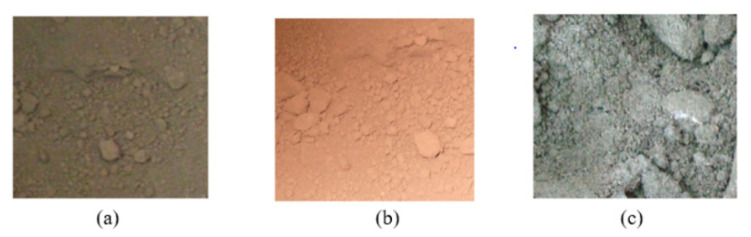
(**a**) Austenitic and (**b**) ferritic stainless steel and (**c**) slag.

**Figure 2 materials-15-05980-f002:**
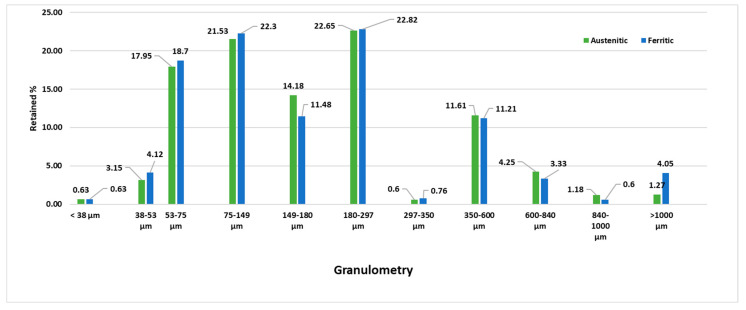
Granulometry of fume dust.

**Figure 3 materials-15-05980-f003:**
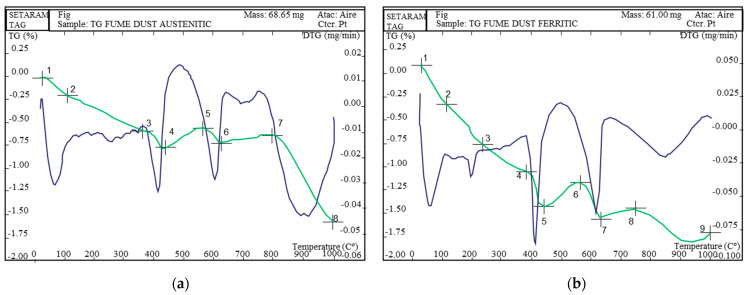
Thermogravimetry of austenitic (**a**) and ferritic (**b**) fume dust.

**Figure 4 materials-15-05980-f004:**
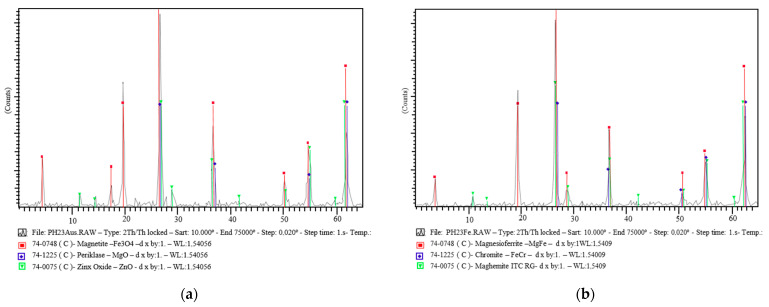
Diffractograms of austenitic (**a**) and ferritic (**b**) flue gas powder.

**Figure 5 materials-15-05980-f005:**
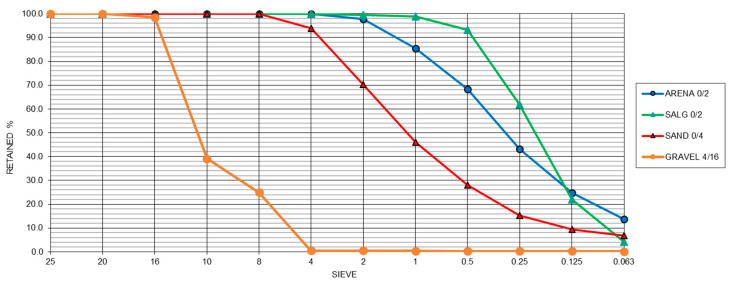
Sand and slag granulometry.

**Figure 6 materials-15-05980-f006:**
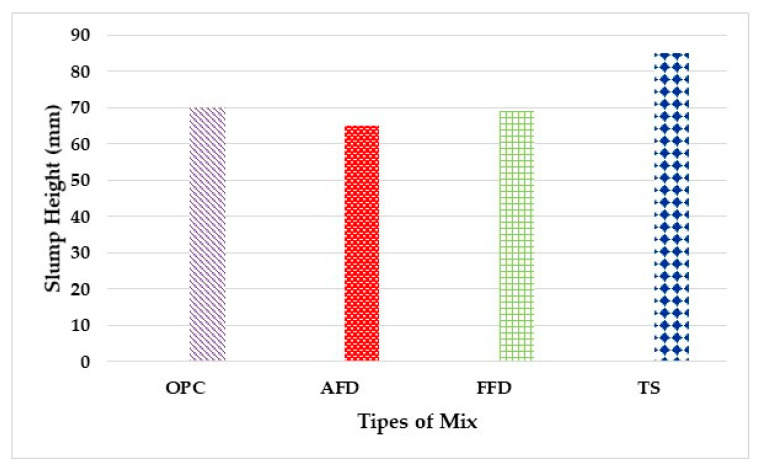
Workability of fresh concrete.

**Figure 7 materials-15-05980-f007:**
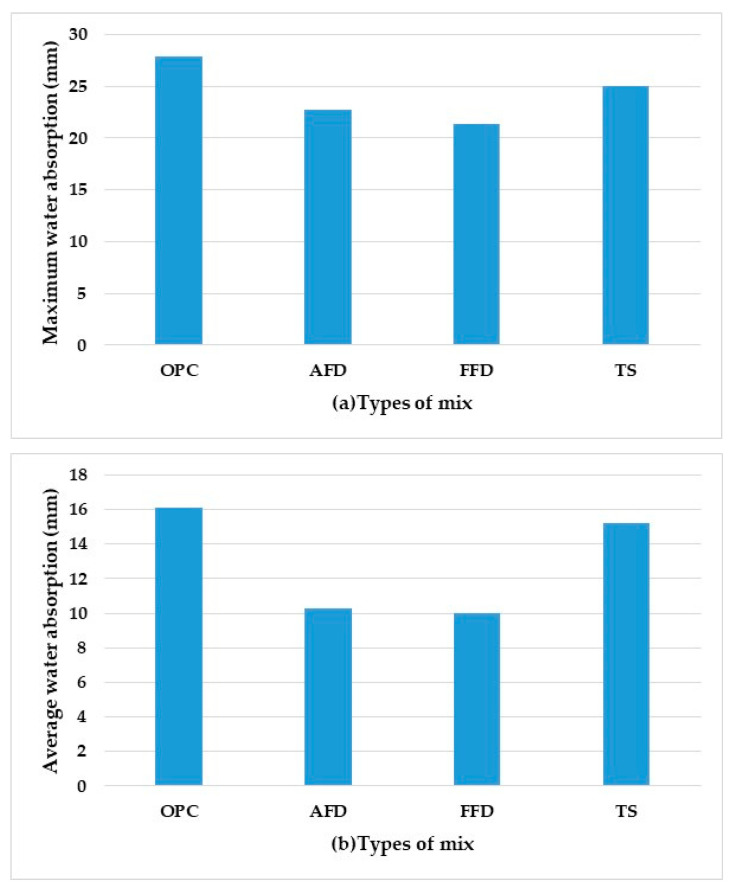
(**a**) Maximum water absorption. (**b**) Average water of different mix types.

**Figure 8 materials-15-05980-f008:**
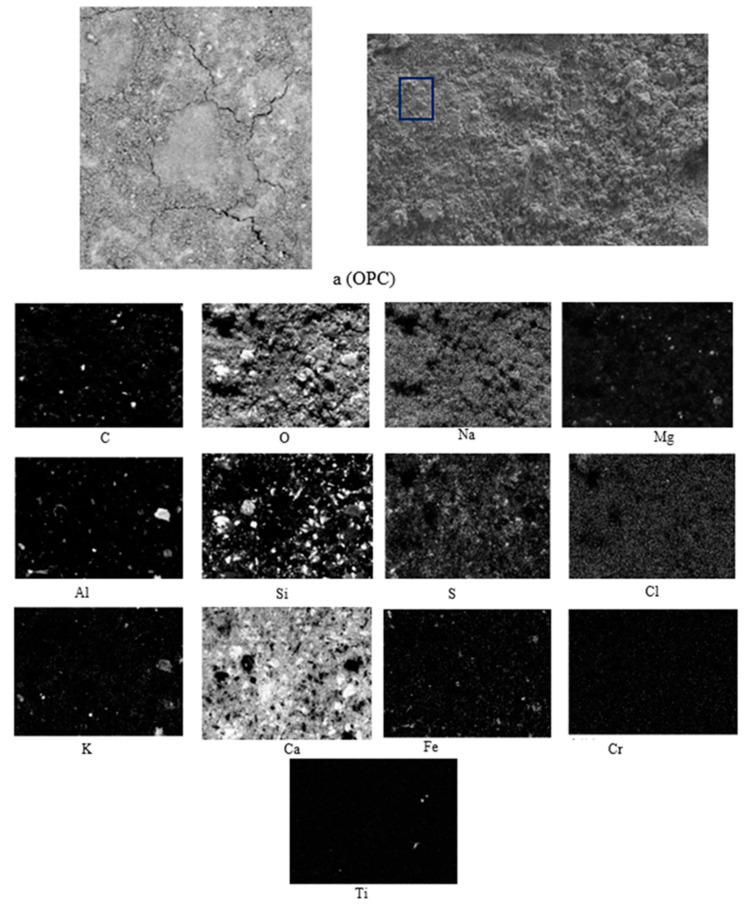
SEM-BSE of OPC and EDS map of element distribution in TRAD mortar sample. (a) Sampling.

**Figure 9 materials-15-05980-f009:**
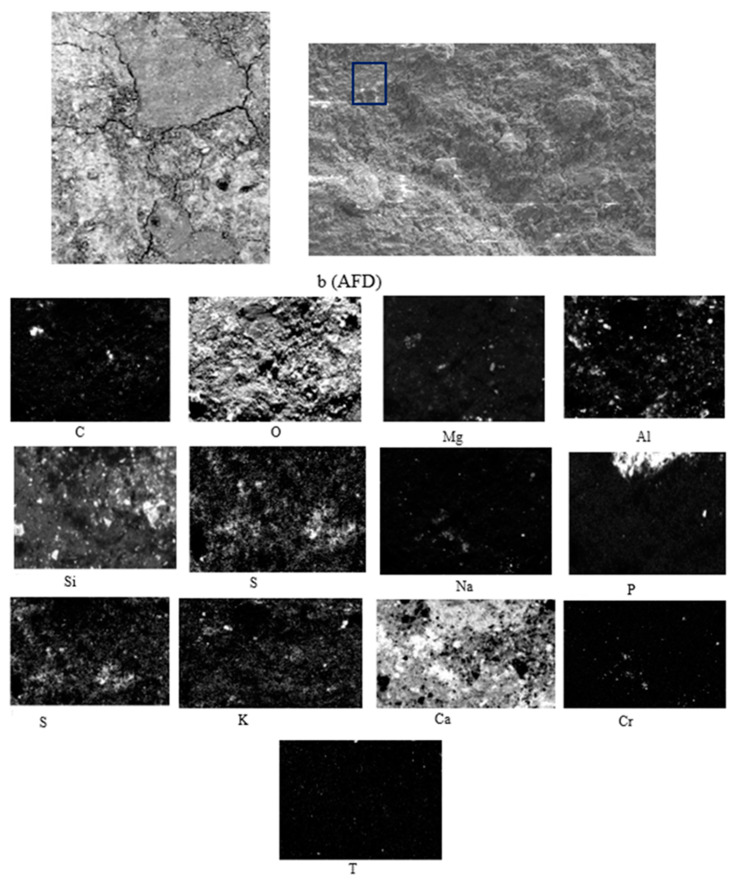
SEM-BSE of AFD and EDS map of element distribution in TRAD mortar sample. (b) Sampling.

**Figure 10 materials-15-05980-f010:**
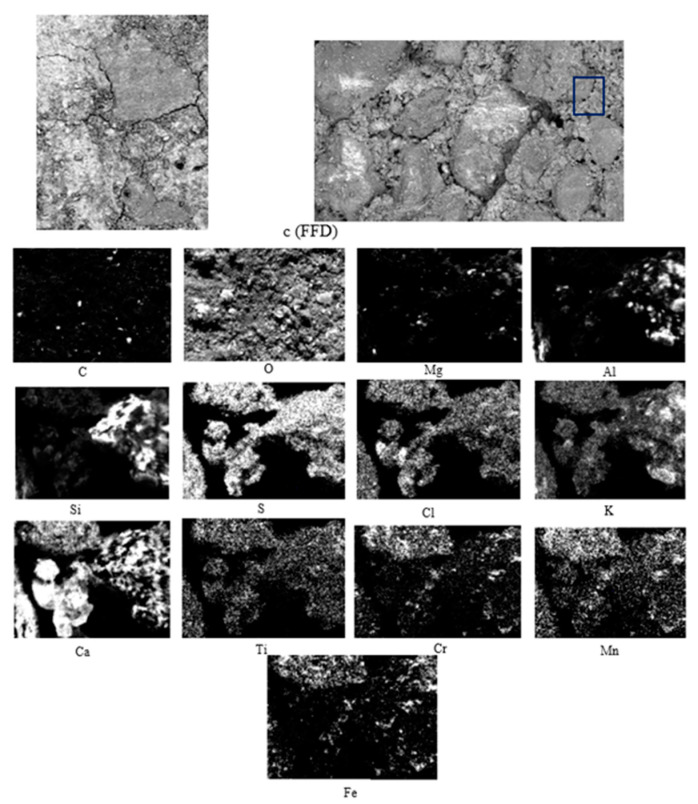
SEM-BSE of FFD and EDS map of element distribution in TRAD mortar sample. (c) Sampling.

**Figure 11 materials-15-05980-f011:**
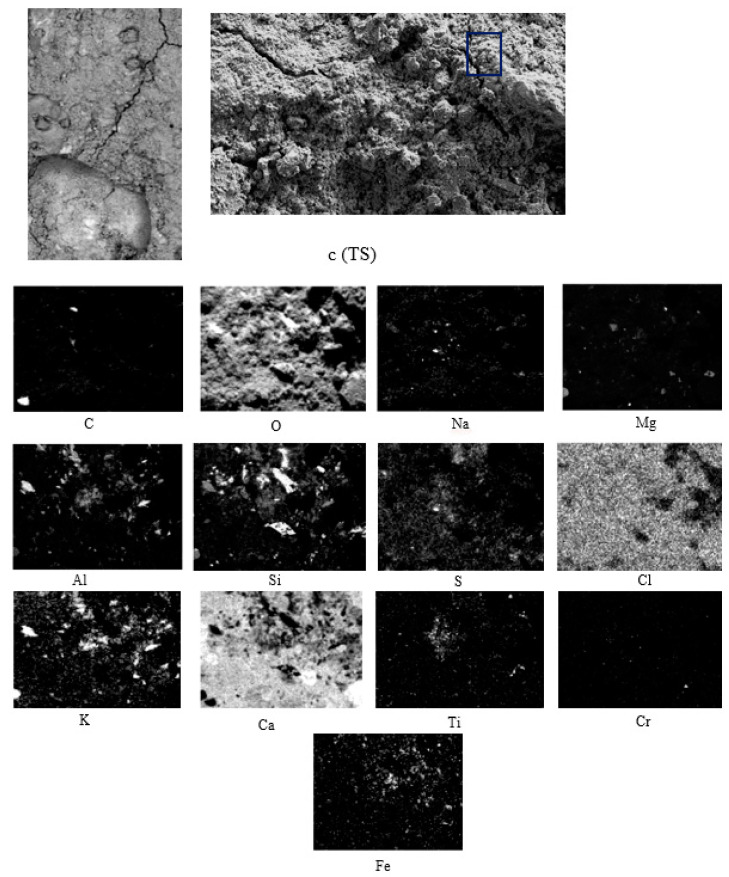
SEM-BSE of TS and EDS map of element distribution in TRAD mortar sample. (c) Sampling.

**Figure 12 materials-15-05980-f012:**
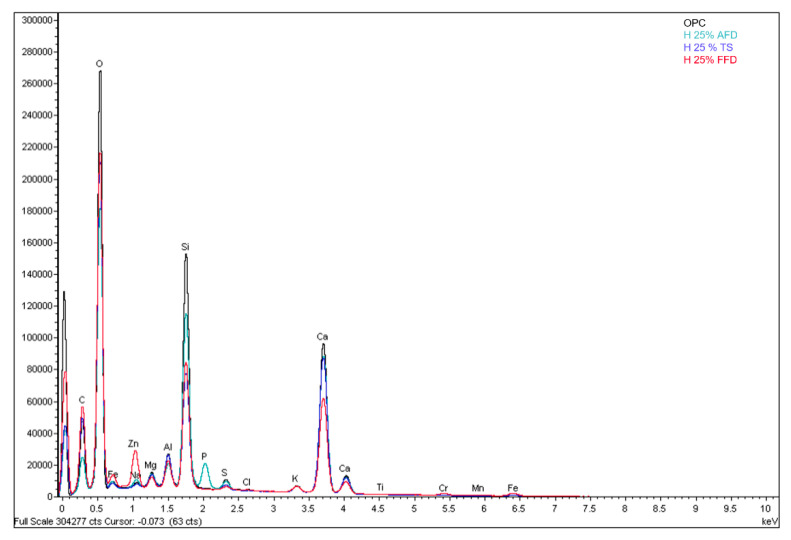
EDS spectra obtained from the surface of OPC, AFD, FFD, and TS mortar samples.

**Table 1 materials-15-05980-t001:** Chemical composition of slag.

Element Oxide	Chemical Composition (%)
CaO	35–60
MgO	4–12
SiO_2_	27–37
Al_2_O_3_	2–6
Cr_2_O_3_	1–8
MnO	1–3
FeO	0.5–4
TiO_2_	1–2
P_2_O_5_	0–0.02

**Table 2 materials-15-05980-t002:** Typical composition of metals in flue dust.

	Austenitic		Ferritic	
	% Metal	% Oxide	% Metal	% Oxide
Zinc	7.75	9.64	4.93	6.13
Lead	0.65	0.60	0.77	0.83
Nickel	2.43	3.18	2.48	3.25
Silicon	3.53	7.55	3.61	7.72
Manganese	3.12	3.99	3.47	4.44
Iron	27.01	38.62	26.26	37.55
Chromium	11.68	17.17	13.88	20.40
Magnesium	2.92	4.85	3.29	5.56
Titanium	0.10	0.17	0.11	0.18
Aluminium	0.43	0.81	0.45	0.85
Calcium	8.06	11.28	6.84	9.58
Tin	0.02	0.02	0.02	0.02
Molybdenum	0.38	0.57	0.20	0.30
Phosphorus	0.02	0.05	0.02	0.05
Copper	0.24	0.30	0.28	0.36
Cadmium	0.24	0.28	0.08	0.10
Sodium	0.70	0.95	0.73	0.98
Potassium	0.80	0.97	1.00	1.20
Chloride	0.62		0.68	
Fluoride	0.17		0.06	
Carbon	0.44		0.33	
Sulfur	0.28		0.30	
Arsenic	0.003	0.005	0.004	0.006
Nitrogen	0.069		0.053	

**Table 3 materials-15-05980-t003:** Organisation of experimental kneading model.

		Binder			Aggregates			
Mix	Water (w/c ratio)	Cement Dosage kg/m^3^	Fume Dust and Slag %	Additive %	Dosage kg/m^3^	Fine Sand 0–2% (mm)	Sand 0–4% (mm)	Gravel 4–16% (mm)
OPC	0.5	325	0%	1.2%	2033.8	15%	50%	50%
AFD		81.25	25% of AFD					
FFD			25% of FFD					
TS			25% of TS					

**Table 4 materials-15-05980-t004:** Compressive strength.

Type of Concrete	f_c_ (MPa) 7 Days	f_c_ (MPa) 28 Days	f_c_ (MPa) 90 Days	E (GPa) 28 Days
OPC	36.8	55.7	58.3	34.21
AFD	48.2	62.6	66.9	40.6
FFD	48.23	67.8	68.5	42.3
TS	31.1	42.2	43.6	28.6

**Table 5 materials-15-05980-t005:** Flexural strength.

Type of Concrete	Flexural Strength (Mpa)
OPC	8.84
AFD	14.67
FFD	14.02
TS	14.44

**Table 6 materials-15-05980-t006:** Analysis of leachate concentration (ppm).

Components Test Tubes			AFD				FFD				TS				OPC	
Leaching			1ª	2ª	1ª	2ª	1ª	2ª	1ª	2ª	1ª	2ª	1ª	2ª	1ª	2ª
Specimen Weight (gr)			362.41		360.35		365.95		365.05		360.15		352.1		353.26	
Detection limits (ppm)	0.034	As	<LD	<LD	0.035	<LD	<LD	<LD	<LD	<LD	<LD	<LD	<LD	<LD	<LD	<LD
	0.1	Ca	1.77	1.08	1.20	1.09	1.59	0.90	1.32	1.13	1.88	1.48	1.72	1.25	1.56	1.76
	0.004	Cr _Total_	0.042	0.025	0.053	0.030	0.045	0.057	0.037	0.034	0.006	0.013	0.008	0.046	0.007	0.023
	0.003	Cu	<LD	<LD	<LD	<LD	<LD	<LD	0.0221	<LD	<LD	<LD	<LD	<LD	<LD	<LD
	0.042	Fe	0.071	<LD	0.079	<LD	0.062	<LD	0.070	<LD	<LD	<LD	0.054	<LD	<LD	<LD
	0.001	Mn	0.001	<LD	0.013	0.001	0.001	<LD	<LD	<LD	<LD	<LD	<LD	<LD	<LD	<LD
	0.09	SO_4−_	10.09	2.78	11.59	3.16	6.91	2.17	4.68	2.05	9.00	5.98	11.13	5.29	17.90	14.15
	0.002	Zn	0.014	<LD	0.010	0.005	0.016	0.002	0.019	0.007	0.010	0.003	0.010	0.013	0.014	<LD
	1.9	Na	24.15	10.22	25.45	10.30	22.51	10.38	19.19	8.37	13.52	6.29	15.49	7.90	16.81	7.15
	0.014	Al	0.084	0.066	0.104	0.072	0.095	0.066	0.090	0.053	0.110	0.048	0.082	0.079	0.071	0.042
	0.056	K	48.46	22.88	52.11	22.80	37.27	18.62	28.52	14.91	23.60	14.59	31.75	18.46	28.15	17.87
	0.025	Si	2.48	1.89	2.63	2.03	2.18	1.76	2.03	1.72	2.88	2.09	3.59	2.40	2.75	1.90
TDS (ppm)			117.53	63.67	124.13	63.84	101.00	57.46	82.36	48.16	72.05	44.57	82.99	52.21	82.59	54.09
Conductivity (µS/cm)			215.0	114.4	225.0	114.7	179.9	103.5	147.2	87.2	129.1	80.9	148.3	94.3	147.6	97.6
pH			9.300	8.620	9.370	8.730	9.250	8.680	9.140	8.320	8.960	8.220	9.180	8.530	8.990	8.160

## Data Availability

Not applicable.

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
