# Peer review of "Development in Sustainable Concrete with the Replacement of Fume Dust and Slag from the Steel Industry"

_materials, 2022, doi:10.3390/ma15175980_

Round 1

Reviewer 1 Report

Comments on “Mechanical characterization of concrete with substitution of slag and fume dust from stainless steels”

This paper investigated the substitution of cement with slag and fume dust in the manufacture of concrete and its effects on the mechanical properties. The research subject could be interesting for the readers. However, the information presented in this paper is similar to the paper published in Construction and Building Materials (https://doi.org/10.1016/j.conbuildmat.2019.01.166), which makes this paper looks like simply repeating previous results. Therefore,  the innovation of this paper has not been highlighted. Moreover, the academical writing of this paper should be significantly improved, in which the logical links to presents the information should be focused. Therefore, based on the current format, it cannot be accepted.

General Comments

1)    Title: The title is too slipper; the keywords is confusing.

2)    Abstract: The chemical composition of some steel waste suggests that there are potentially appropriate substances for reusePlease specify the why it can be reused and used in which area? Too much background information is included in the abstract. The aims, methodology and results should be expanded. The innovation of this manuscript has not been clearly presented.

3)    Keywords: stainless steel fume dust and slag is not a valid keyword.

4)    Introduction: I do not know the logic link of Second paragraph, why it is here and what the authors want to express. Similar to the later paragraphs. The authors should introduce the relative background information and then move to the research gap generated from the authors after the detailed literature review, after which comes the aim and research question. The authors simply introduced some irrelative information, whereas the mechanical characterisation, the key information about this manuscript, is ignored. Therefore, the literature review parts in introduction should be significantly improved and the innovation about the paper should be highlighted.

5)    Experiment: Materials: The resource, chemical composition and physical properties for all important materials should be included, in which section 2.1 and 2.2 are suggested to be combined. Lines 163-183, too much general information is included, which should be concentrated and only keep the relative information. Remember this is experiment part not results part. Section 3 should be part of the experimental section, which is also suggested to be combined. Fig 6 is not necessary. 3.1. explain why the authors select the slump between 30-60? Provide the details about the mechanical test.

6)    Results. How the author test water adsorption? Please specify in previous section. Table 4 (and Table 5), provide the standard derivation, and why only providing 28-day’s Modulus? SEM, the resolution of image is too low to identify. Figure 12, how the author concluded “crystals of different shapes and size…”. I could not see from the images. Table 6, why many marks about correction is remained in the manuscript? What is 1a  2a 3a and 4a? The Table 6 should be re-organised based on the academic regulation. The analysis on the mechanism of leaching should be provided.

7)    Conclusion. How the authors concludes “permeability” results?

Author Response

Reviewer #1:

This paper investigated the substitution of cement with slag and fume dust in the manufacture of concrete and its effects on the mechanical properties. The research subject could be interesting for the readers. However, the information presented in this paper is similar to the paper published in Construction and Building Materials (https://doi.org/10.1016/j.conbuildmat.2019.01.166), which makes this paper looks like simply repeating previous results. Therefore,  the innovation of this paper has not been highlighted. Moreover, the academical writing of this paper should be significantly improved, in which the logical links to presents the information should be focused. Therefore, based on the current format, it cannot be accepted.

Thanks for the observation. Actually, the paper published in "Construction and building materials" reports the results obtained from specimens with only "addition" (5, 10 and 15%) of ferritic fume dust to the mixture with no changes in the quantities of the other constituents. Conversely, the present study investigates the consequences of "replacing" cement with other by-products employed as alternative binders, which is one of the relevant novelties of this paper.

 General Comments

  • Title: The title is too slipper; the keywords is confusing.

Reply

Thanks for the observation. The title has been modified. “Development in sustainable concrete with replacement of fume dust and slag from the production of stainless steel”

  • Abstract: “The chemical composition of some steel waste suggests that there are potentially appropriate substances for reuse ”Please specify the why it can be reused and used in which area? Too much background information is included in the abstract. The aims, methodology and results should be expanded. The innovation of this manuscript has not been clearly presented.

Reply

Thank you for your suggestion. Abstract has been completely rewritten.

  • Keywords: stainless steel fume dust and slag is not a valid keyword.

Reply

Thank you for your suggestion. The list of keywords has been modified and the reviewer's comments has been taken into account.

  • Introduction: I do not know the logic link of Second paragraph, why it is here and what the authors want to express. Similar to the later paragraphs. The authors should introduce the relative background information and then move to the research gap generated from the authors after the detailed literature review, after which comes the aim and research question. The authors simply introduced some irrelative information, whereas the mechanical characterisation, the key information about this manuscript, is ignored. Therefore, the literature review parts in introduction should be significantly improved and the innovation about the paper should be highlighted.

Reply

Thank you for your comment. The introduction section has been made more concise and a more straightforward logic path has been followed. Specifically, the main paragraphs of the section deal with the following aspects:

  • a general introduction of waste production in industrial processes;
  • mention to the stainless steel production and definition of the two by-product of relevance for the present study
  • aims and objectives of the present study, whose main novelties are: i) the use of by-products coming from the production of stainless steel;
  •  
  • Substitution of stainless steel residues in concrete, since there are a large number of batches with blast furnace slag and carbon steel electric arc furnace by substitution of aggregates but by substitution of cement and in proportions of 25% we have not found Regarding the methodology used, the characterization of the dust and slag of stainless steel fumes was studied and subsequently different mechanical and durability tests were carried out to see if it was viable. Based on this, the work was structured.

  • Experiment: Materials: The resource, chemical composition and physical properties for all important materials should be included, in which section 2.1 and 2.2 are suggested to be combined. Lines 163-183, too much general information is included, which should be concentrated and only keep the relative information. Remember this is experiment part not results part. Section 3 should be part of the experimental section, which is also suggested to be combined. Fig 6 is not necessary. 3.1. explain why the authors select the slump between 30-60? Provide the details about the mechanical test.

Reply

Thank you for your comment. Sections 2.1 and 2.2 have been merged. Lines 163 to 183 have been reduced according to the reviewer’s comment. Section 3 has been placed within the experimental section. Figure 6 has been removed. The slump test has been developed according to regulations.

  • How the author test water adsorption? Please specify in previous section. Table 4 (and Table 5), provide the standard derivation, and why only providing 28-day’s Modulus? SEM, the resolution of image is too low to identify. Figure 12, how the author concluded “crystals of different shapes and size…”. I could not see from the images. Table 6, why many marks about correction is remained in the manuscript? What is 1a  2a 3a and 4a? The Table 6 should be re-organised based on the academic regulation. The analysis on the mechanism of leaching should be provided.

Reply

Thanks for your comment. It is true that the essay to perform the test had not been developed. This has been modified and explained in the document. In relation to Young's modulus, as carried out by the UNE-EN 12390-13:2022 test, the specimens used were cylindrical 15x30 cm to measure the value with greater precision, since it reaches approximately 70% of the resistant capacity. at that age. The resolution of the images was provided by the company that supplied the waste with this quality. Markups in manuscript have been corrected. Table 6 has been modified, on the other hand, as discussed in the work, a second leaching test was carried out once the specimen had aged to confirm that there was no type of leaching in the concrete. This would be the development of the test “In the development of the leaching process, the test tube was introduced in 800 mL of distilled water for 24 hours. The test was repeated by immersing the test tube for 24 hours in another 800 mL of water, repeating the described methodology. After the test period, the sample was removed and the leach solution was transferred to a 1000 mL volumetric flask filled with distilled water. Subsequently, the test tubes were artificially aged for 3 months and the same process was carried out”.

  • How the authors concludes “permeability” results?

Reply

Thanks for the correction. It was a mistake to introduce permeability instead of capillarity. The terminology has been amended in the revised manuscript.

Reviewer 2 Report

The paper "Mechanical characterization of concrete with substitution of slag and smoke dust from stainless steel" is interesting, well written and adequate, but it needs some tweaking:

(i) The title has a point, it must be revised!!!

(i) The abstract must be reformulated to be more specific and present quantitative results;

(ii) The introduction could be improved with the addition of more recent work on the topic of study, note that a comprehensive review is required, such as: 10.3390/app11073036; 10.1016/j.cscm.2022.e01255; 10.1016/j.cscm.2022.e00948.

(iii) In my view, the authors addressed few samples of material incorporation, this is a general problem with the paper. Another relevant question, what is the real innovation of this research??? There are numerous roles of this type.

(iv) An experimental flowchart can be added;

(v) Many images and graphics are difficult to read, texts are too small, check them all;

(vi) Table 1 the percentage of CaO and other elements cannot be that comprehensive!

(vii) Table 3 should be fully revised;

(viii) There is no statistical analysis, at least one error bar must be implemented;

(ix) The conclusion is very extensive and should be reformulated, more objective to the readers.

Author Response

Reviewer #2:

The paper "Mechanical characterization of concrete with substitution of slag and smoke dust from stainless steel" is interesting, well written and adequate, but it needs some tweaking: 

  • The title has a point, it must be revised!!!

Reply

Thanks for the observation. The title has been modified, also with the aim to respond to a comment raised by Reviewer #1.

  • The abstract must be reformulated to be more specific and present quantitative results;

Reply

Thank you for your suggestion. The abstract has been rewritten: the comments raised by both Reviewers #1 and 2 have been taken into account.

  • The introduction could be improved with the addition of more recent work on the topic of study, note that a comprehensive review is required, such as: 10.3390/app11073036;10.1016/j.cscm.2022.e01255;10.1016/j.cscm.2022.e00948.

Reply

Thank you for your suggestion. The introduction section has been significantly revised. Moreover, the recommended paper has been added to the reference list.

  • In my view, the authors addressed few samples of material incorporation, this is a general problem with the paper. Another relevant question, what is the real innovation of this research??? There are numerous roles of this type.

Reply

Thank you for your observation. It is considered that the samples are representative based on the data obtained since the tolerance margins have been within what was expected. On the other hand, for each test carried out, as many samples as required by regulations have been used. A paragraph has been introduced with the aim to further highlight that one of the main novelties of the paper is that is refers to by-products obtained from the production of stainless steel, which are much less investigated (than those obtained from common carbon steel) as cement replacement in concrete.

  • An experimental flowchart can be added;

Reply

Thank you for your observation. Due to space constraints we have decided not to add a flow chart: it would certainly be of help to reader, but we believe that all the relevant information are already reported within the text.

  • Many images and graphics are difficult to read, texts are too small, check them all;

Reply

Thank you for your suggestion. All graphics have been modified accordingly.

  • Table 1 the percentage of CaO and other elements cannot be that comprehensive!

Reply

Thank you for your suggestion. Actually, the reported CaO percentage values were obtained from chemical composition tests.

  • Table 3 should be fully revised;

Reply

Thanks for the observation. As requested, Table 3 has been thoroughly revised.

  • There is no statistical analysis, at least one error bar must be implemented;

Reply

Thank you for your suggestion. The tolerance has not been put in the tables, but it is reported within the text. The results obtained are the average of the rupture values with a deviation of ±5 MPa"

  • The conclusion is very extensive and should be reformulated, more objective to the readers.

Reply

Thanks for your suggestion. The conclusions significantly revised, with the aim to make them sharper and more concise.

Reviewer 3 Report

1) Please write some performance values in the abstract, as well as in the conclusions. Now, there is only percentages are written. E.g. "The most outstanding values in 17 the substitutions carried out were around 21% in compression and 25% in bending in terms of increased resistance capacity." but we did not get the baseline, and reading only the abstract or the conclusions, we did not get any performance data, just the percentages of relative improvements. I advise giving some performance values in MPa or GPa both in abstract and conclusions. 

2) In line 49, please explain AOD, before using the abbreviation.

3) In the introduction chapter, I suggest to avoid one sentence long paragraphs.

4) I may suggest merging chapter 2 and 3 into Materials and Methods. 

5) Fig 1. and Fig. 6. caption claims scale 1:1, but there is no visible scale on the pictures. There are two fig. 1 in the paper.  

6) Generally, many figures could be larger to provide better readability. E.g. the second fig. 1's or fig 7's letter colour are grey now and too small. Fig 8 contains two figures, but they are not marked with a) and b) and not described separately in the caption. Please improve all the figures for better readability and aesthetics.

7) How many sample were examined for granularity? Is 250 g sample representative for the whole compound? How did you tested it? Are fig 1. values from a single test or averages of multiple tests? What was the deviation of the tests, if multiple tests were performed?

8) What is the source of Table 1 data? Authors wrote only that "company" provided the data, but the Tables does not show any reference. I also suggest avoiding page breaks during Tables.

9) I suggest describing the thermogravimetric analysis. The results may belongs better into the results chapter. Results shown in Fig. 3. is not analysed in the paper.

10) The diffractometric analysis is also just mentioned and the methodology and devices are not described in the manuscript. The text does not refer to Fig 4. (which is also too small and low res).

11) What is the source of Fig. 5? If it is a result of the authors' then I suggest including the test results in the results chapter, and in the materials and methods chapter, only describing the tests performed by the authors (but in details). 

12) Table 3 is not in edited in the journal's format.

13) The methodology of the tests performed during the research should be described in the materials and methods chapter, not in the results. Therefore I advise adding the descriptions of compression, flexural strength, water absorption, SEM, etc. into the methods chapter. 

14) Please write units after numbers, e.g. line 270, the "deviation is +-5"...

15) Table 4 does not contains the deviations, only the averages. However, the caption of the Table does not mention that the values shown are averages. I suggest including the standard deviation of each test. Please, change decimal commas into decimal points.

16) There is no scale in SEM figures! The methodology and description of scanning electron microscopy is also missing. 

17) Fig 9-13 shows single sample SEM based tests. How representatives are these single tests for the whole sample? Fig 13 show element identification, but also based on single samples. 

18) I advise to outline the novelty and scientific contribution in the conclusions, while also mentioning the limitations of the study. I also advise to put not only relative, but specific key results into the conclusions.

19) Generally, I suggest a large revision and rewriting of the paper in the suggested form, then resubmit it. 

Author Response

Reviewer #3:

  • Please write some performance values in the abstract, as well as in the conclusions. Now, there is only percentages are written. E.g. "The most outstanding values in 17 the substitutions carried out were around 21% in compression and 25% in bending in terms of increased resistance capacity." but we did not get the baseline, and reading only the abstract or the conclusions, we did not get any performance data, just the percentages of relative improvements. I advise giving some performance values in MPa or GPa both in abstract and conclusions. 

Reply

Thanks for your suggestion. However, the authors consider that the data are in the tables of the article and, therefore, they believe it more convenient to give percentage values to be able to take into account the tolerances of each test.

  • In line 49, please explain AOD, before using the abbreviation.

Reply

Thanks for your observation. Due to the suggestions of the other reviewers, it has been decided to eliminate the paragraph where the abbreviation appears.

  • In the introduction chapter, I suggest to avoid one sentence long paragraphs.

Reply

Thanks for your observation. Fixed based on suggestions

  • I may suggest merging chapter 2 and 3 into Materials and Methods. 

Reply

Thanks for your observation. The dots have been connected based on the suggestion

  • Fig 1. and Fig. 6. caption claims scale 1:1, but there is no visible scale on the pictures. There are two fig. 1 in the paper.  

Reply

Thanks for your observation. It is true that the scales have been eliminated.

  • Generally, many figures could be larger to provide better readability. E.g. the second fig. 1's or fig 7's letter colour are grey now and too small. Fig 8 contains two figures, but they are not marked with a) and b) and not described separately in the caption. Please improve all the figures for better readability and aesthetics.

Reply

Thanks for your observation. The scales of the legends have been changed, as well as the colour, putting them in bold for better reading. On the other hand, Figure 8 has been described in two separate sections.

  • How many sample were examined for granularity? Is 250 g sample representative for the whole compound? How did you tested it? Are fig 1. values from a single test or averages of multiple tests? What was the deviation of the tests, if multiple tests were performed?

Reply

Thank you for your observation. For the study of the granulometry of the aggregates and the slag, the UNE-EN 933-2 standard was used. Based on these data, the test was carried out. Conversely, the 250 g fume powders are representative since this type of residue is very homogeneous and, due to its low density, this amount of sample it is sufficient, since its granulometry is measured in specific instrumentation based on its fineness not being able to be done with conventional sieving. Only one test was carried out since, as indicated above, it is a very homogeneous material.

  • What is the source of Table 1 data? Authors wrote only that "company" provided the data, but the Tables does not show any reference. I also suggest avoiding page breaks during Tables.

Reply

Thank you for your observation. Data were supplied by Acerinox SA, which is mentioned within the "Acknwledgement" section.

  • I suggest describing the thermogravimetric analysis. The results may belongs better into the results chapter. Results shown in Fig. 3. is not analysed in the paper.

Reply

Thank you for your observation. These tests are characteristics of the constituents. Therefore, we decided to report them in the "Materials and method" section. The "Results" section is devoted to report the outcome of the experimental tests on the concrete mixtures under consideration in this study.

  • The diffractometric analysis is also just mentioned and the methodology and devices are not described in the manuscript. The text does not refer to Fig 4. (which is also too small and low res).

Reply

Thank you for your observation. These are data provided by the test instruments and it is not possible to change the type of letter offered by it. However, both figures 3 and 4 have been stretched and enlarged, which should make them more easily readable. Moreover, a reference to Figure 4 has been added in the text.

  • What is the source of Fig. 5? If it is a result of the authors' then I suggest including the test results in the results chapter, and in the materials and methods chapter, only describing the tests performed by the authors (but in details). 

Reply

Thank you for your observation. The figure refer to the charactersisation of concrete constituents and, thus, it has been reported within the "Materials and Methods" section.

  • Table 3 is not in edited in the journal's format.

Reply

Thank you for your observation. It is true the format has been corrected.

  • The methodology of the tests performed during the research should be described in the materials and methods chapter, not in the results. Therefore I advise adding the descriptions of compression, flexural strength, water absorption, SEM, etc. into the methods chapter. 

Reply

Thank you for your observation. This has been fixed based on feedback from all reviewers

  • Please write units after numbers, e.g. line 270, the "deviation is +-5"...

Reply

Thank you for your observation. This information has already been corrected.

  • Table 4 does not contains the deviations, only the averages. However, the caption of the Table does not mention that the values shown are averages. I suggest including the standard deviation of each test. Please, change decimal commas into decimal points.

Reply

Thank you for your comment. Actually, the test tolerance is reported at the beginning of the paragraph. The deviations are not reported also within the table for the sake of readability.

  • There is no scale in SEM figures! The methodology and description of scanning electron microscopy is also missing. 

Reply

Thank you for your comment. The methodology of machinery that offers the data directly has been ignored, since explaining the operation of each one of them would considerably increase the number of pages of the paper.

  • Fig 9-13 shows single sample SEM based tests. How representatives are these single tests for the whole sample? Fig 13 show element identification, but also based on single samples. 

Reply

Thanks for your comment. They are the samples made to each specimen for each dosage described.

  • I advise to outline the novelty and scientific contribution in the conclusions, while also mentioning the limitations of the study. I also advise to put not only relative, but specific key results into the conclusions.

Reply

Thank you for your observation. This has been fixed based on comments from all reviewers.

  • Generally, I suggest a large revision and rewriting of the paper in the suggested form, then resubmit it. 

Reply

Thanks for your comments. They have been of great help in revising the original manuscript, as several parts of the manuscript have been actually rewritten. We hope the current version will find the reviewer’s approval.

Round 2

Reviewer 1 Report

Most of my previous comments have been carefully addressed by the authors. The paper can meet the standard for publishing in the journal after some minor correction on academic writing.

Author Response

Thank you

Reviewer 2 Report

ok.

Author Response

Thank you.

Reviewer 3 Report

Authors did not adressed several comments of the reviewer, and they also left obvious editing errors in the article.

1) Only percentages are present in the abstract which does not give a clue of the baseline or the improved resistance capacity. 

2) Fig 1. and Fig. 6. caption claims scale 1:1, but there is no visible scale on the pictures. Delete it, if it is not scaled... There are two fig. 1 in the paper, correct it next time you submit the paper somewhere. 

3) Generally, many figures could be larger to provide better readability. E.g. the second fig. 1's or fig 7's letter colour are grey now and too small. Fig 8 contains two figures, but they are still not marked with a) and b). Could you please check the journal's word template, how to mark a figure?! Please improve all the figures for better readability and aesthetics. Still unequeal quality and aesthetics.

4) Standards are not cited in the list of references, e.g. EN 933-2.

5) Reporting a test result based on a single measurement is unprofessional, even if the sample is considered by the eye "very homogeneous". 

6) If a data in the manuscrit is from a source, please mention and cite it in the text. Not only that "the material is provided by xy", but Table 1 and Table 2 has a reference, it is not your measured values. It should be cited. If these values you report are available online, then simply cite them and don't put it in your paper. 

7) The diffractometric analysis is also just mentioned and the methodology and devices are not described in the manuscript, and the text still does not refer to Fig 4. anywhere.

8) The methodology of the tests performed during the research should be described in the materials and methods chapter, not in the results. It is not corrected. Therefore I still suggest adding the descriptions of compression, flexural strength, water absorption, SEM, etc. into the methods chapter. I also suggest using the same order in materials and methods and results chapter to present the experiments.

9) Table 4 does not contains the deviations, only the averages. However, the caption of the Table does not mention that the values shown are averages. If it is a scientific paper, then stating that all of the results are within +- 5 MPa is very unprofessional. That is around +-7.3-16.6% difference based on the reported results. 

10) There is no scale in SEM figures! The methodology and description of scanning electron microscopy is still missing.

11) Fig 9-13 shows single sample SEM based tests. How representatives are these single tests for the whole sample? Fig 13 show element identification, but also based on single samples.

12) Since many suggestion and question were not addressed or answered, I again suggest reject.

Author Response

Reviewer #3:

Authors did not adressed several comments of the reviewer, and they also left obvious editing errors in the article.

  1. Only percentages are present in the abstract which does not give a clue of the baseline or the improved resistance capacity.

Reply

Thanks for your observation. The introduction has been modified based on all the revisions, making it as homogeneous as possible based on the different opinions.

  1. Fig 1. and Fig. 6. caption claims scale 1:1, but there is no visible scale on the pictures. Delete it, if it is not scaled... There are two fig. 1 in the paper, correct it next time you submit the paper somewhere.

Reply

Thanks for your observation. The scale and figure 6 have been removed.

  1. Generally, many figures could be larger to provide better readability. E.g. the second fig. 1's or fig 7's letter colour are grey now and too small. Fig 8 contains two figures, but they are still not marked with a) and b). Could you please check the journal's word template, how to mark a figure?! Please improve all the figures for better readability and aesthetics. Still unequeal quality and aesthetics.

Reply

Thanks for your observation. The color of the letters has been had and increased and changed and they have become larger. As for figure 8, it would only be the one corresponding to “a)” since they are elements of the figure that are part of a small section, for this reason it has been considered convenient not to make more subdivisions so as not to lead to error, avoiding putting “b )”.

  1. Standards are not cited in the list of references, e.g. EN 933-2.

Reply

Thanks for your observation. It has been modified and inserted into the bibliography

  1. Reporting a test result based on a single measurement is unprofessional, even if the sample is considered by the eye "very homogeneous".

Reply

Thanks for your observation. As a general rule, in this type of study, a representative sample of the material to be used is extracted, making the study of said sample because the tests are very expensive.

  1. If a data in the manuscrit is from a source, please mention and cite it in the text. Not only that "the material is provided by xy", but Table 1 and Table 2 has a reference, it is not your measured values. It should be cited. If these values you report are available online, then simply cite them and don't put it in your paper.

Reply

Thanks for your observation. The data on the composition of the materials, as in the previous question, is extracted from a sample that has been studied later. Acerinox carried out the test for said test and an average of each of its components is studied. Due to Industrial Secret policies, the company only authorized the authors to show said data since there is a confidentiality agreement with it.

  1. The diffractometric analysis is also just mentioned and the methodology and devices are not described in the manuscript, and the text still does not refer to Fig 4. anywhere.

Reply

Thanks for your commentary. The error has been corrected and the one corresponding to figure 4 has been introduced.

  1. The methodology of the tests performed during the research should be described in the materials and methods chapter, not in the results. It is not corrected. Therefore I still suggest adding the descriptions of compression, flexural strength, water absorption, SEM, etc. into the methods chapter. I also suggest using the same order in materials and methods and results chapter to present the experiments.

Reply

Thanks for your commentary. The authors have adapted to the comments of the rest of the reviewers and mention has been made of the development of the trials in the regulations that are used for them.

  1. Table 4 does not contains the deviations, only the averages. However, the caption of the Table does not mention that the values shown are averages. If it is a scientific paper, then stating that all of the results are within +- 5 MPa is very unprofessional. That is around +-7.3-16.6% difference based on the reported results.

Reply

Thanks for your observation. No table of deviations has been introduced,  the final average is set by breaking two specimens for each of the ages studied, according to the regulations, and if the deviation is greater than 5% (this was not our case), a third specimen is broken by introducing the proportional data in the test. Therefore, only those breakage data according to regulations are shown in the table.

  1. There is no scale in SEM figures! The methodology and description of scanning electron microscopy is still missing.

Reply

Thanks for your observation. As for the methodology and description of the SEM microscope test, this equipment is very expensive and is not found at the University, and the tests are carried out at Acerinox. As I have mentioned before, I signed a confidentiality agreement with Acerinox and it only allows me to show the results and therefore carry out the analysis based on them.

  1. Fig 9-13 shows single sample SEM based tests. How representatives are these single tests for the whole sample? Fig 13 show element identification, but also based on single samples.

Reply

Thanks for your observation. As in the previous answers, these tests are very expensive and a representative sample was studied.

  1. Since many suggestion and question were not addressed or answered, I again suggest reject.

Reply

The authors see the reviewer’s point and hope that the new revision of the manuscript (and the responses reported above) sound sufficiently convincing and the manuscript can finally find the approval of Reviewer #3, as it was already the case with both Reviewers #1 and #2.

Round 3

Reviewer 3 Report

- Fig. 2. and Fig. 5. use too small text and are barely readable. Numbers use commas instead of decimal points.

- Chapter 2.2 shall be "Morphology". 

- Authors should expand the description of the thermogravimetric analysis they made (from line 132). The diffractometric analysis is also not described in the manuscript and should be extended (from line 139). 

- Fig. 3. and Fig. 4. are too small and not visible correctly. Both figures contain two graphs. Therefore they should be marked with a) and b) and described separately in the caption.

- Equation in line 149 shall be numbered.

- Table 3. is not formatted correctly. kg is not with capital K. 

- Water absorption, unaxial compression test, flexural strength test, scanning electron microscopy (SEM) with energy dispersive x-ray spectroscopy (EDS), and leachate analysis are not described in the chapter which is called "experimental work". Only the workability test is mentioned in that chapter, which should be called materials and methods and describe the tested materials and the experiments performed. The paper shall be reconsidered after this is done during a major revision. 

- Since the authors did not give any standard deviation in their manuscript and performed single sample tests in multiple experiments only, the lack of a multisample study shall be mentioned as a limitation in their manuscript. 

- Author contributions, Funding, Data availability statement, etc. is missing.

Author Response

Pdf file is included
